# Nest architecture and colony composition in two populations of *Ectatomma ruidum* sp. 2 (*E. ruidum* species complex) in southwestern Colombia

Carlos Santamaría[1]*, Inge Armbrecht[1], Jean-Paul Lachaud[2,3]*

**1** Departamento de Biología, Universidad del Valle, Cali, Colombia, **2** Departamento de Conservación de la Biodiversidad, El Colegio de la Frontera Sur (ECOSUR), Chetumal, Quintana Roo, Mexico, **3** Centre de Recherches sur la Cognition Animale (CRCA), Centre de Biologie Intégrative (CBI), Université de Toulouse, CNRS, UPS, Toulouse, France

\* jean-paul.lachaud@univ-tlse3.fr (JPL); csantave@gmail.com (CS)

**Data Availability Statement:** All relevant data are within the paper and its Supporting Information files.

## Abstract

Nest architecture plays a fundamental role in the adaptation of ants to their habitat, favoring the action of economically important species. *Ectatomma ruidum* sp. 2 (*ruidum* species complex) is a biological control agent in Neotropical agroecosystems, exhibiting high bioturbation impact due to high nest densities. The architecture and composition of 152 nests were studied in two Andean populations of southwestern Colombia, 24 of them being cast using the paraffin wax technique. Nest entrance was a single, circular, 4 mm hole at ground level, without any special external structure, connected to a single vertical tunnel communicating with successive half ellipsoidal chambers. Nests were extremely shallow (depth range: 28.7–35.4 cm), with an average of six chambers and an overall volume of 92.2 cm$^3$ per nest. The deeper the chamber, the smaller its volume. Nest building was independent of plants or roots, and no surface or underground physical connections were found between neighboring nests. Few nests possessed a queen, and neither ergatoids nor microgynes were recorded. Despite significant interactions between localities and the number of both males and workers, queen presence had an overall highly positive effect on the number of workers and larvae and a negative one on the number of gynes. Overall, the studied Colombian populations of *E. ruidum* sp. 2 retained the simple nest structure described for other species of this species complex and for colonies of the same species from other geographical areas, though they constrasted in their extreme shallowness. Our data suggest that *E. ruidum* sp. 2, at the local level, does not follow the usual monodomic pattern of this species with facultative polygyny but, rather, has a polydomic pattern with monogyny, perhaps related to the extreme shallowness of the nests due to soil structure, which could significantly enhance the queen's reproductive inhibition previously reported for this species.

**Funding:** Funds to CS were provided by COLCIENCIAS (National Basic Sciences Program, Code 110656933821, Contract RC. No. 0648-2013), by the 'Vicerrectoría de Investigaciones de la Universidad del Valle' ('International Visibility' call, code CI 71000), and by the founding of the research projects (2015's call #727) for National PhD students by COLCIENCIAS and the Postgraduate Department of Biological Sciences of the Universidad del Valle. The funders had no role in study design, data collection and analysis, decision to publish, or preparation of the manuscript.

**Competing interests:** The authors have declared that no competing interests exist.

# Introduction

Ants are considered ecosystem engineers [1] and their functional role on soil dynamics is directly related to both their key impact as soil modifiers and their considerable bioturbation activity through nest-building [2–6]. Nest architecture and nesting material choice play a fundamental role in the adaptation of ant colonies to their habitat and their survival through changes in soil physical properties and the resulting effects on both flooding resistance and thermal and humidity regulation [3, 7]. In Africa, for example, the workers of *Brachyponera sennaarensis* use the hydrophobic properties of the surface sand, lining the interior of the galleries with this dry sand to prevent flooding [8]. On the other hand, humidity and temperature conditions can vary according to both the location of the nest and the insulation properties of the nest material [9, 10] but also according to the depth of the chambers. In various species, adults position themselves within the nest according to a temperature gradient according to their physiological state and age [11, 12]. Similarly, immature individuals are distributed in the chambers according to their developmental stage and the thermal requirement ensuring ideal conditions for their development [11, 13].

Numerous works have highlighted that ant nest architecture is species-specific and influences not only the development of the colonies, their composition, and their spatial structuration [12, 14–18] but also their collective behavior [18, 19]. Nest architecture has been described with some details for a few species of the ectatommine Neotropical ant genus *Ectatomma* (*E. opaciventre* [20], *E. planidens* [21] (mistakenly identified as *E. edentatum* [22]), *E. brunneum* [23, 24], and *E. vizottoi* [25]), but a detailed account of the population found within each nest chamber has never been included. Furthermore, for *E. ruidum* and *E. tuberculatum*, the two most common and important species of the genus due to their economic impact as biological control agents [26–28], much of the information concerning their nest architecture has been obtained indirectly, with only the description of a few nests and a lack of information concerning the volume of the chambers and the distribution of the individuals among them [26, 29, 30].

*Ectatomma ruidum* sp. 2 belongs to the *E. ruidum* species complex, which currently consists of at least five derived morphospecies. These are *Ectatomma ruidum* sp. 1, 2, 3–4, 5 and a possible 2x3 hybrid [31–34]. In terms of habitat, this species complex is widely distributed, occupying agricultural systems, savannas, and forests from sea level to an altitude of 1500–1600 m [26] and up to 2200 m [35]. Its latitudinal distribution in the Americas ranges from the states of Jalisco and Nayarit in Mexico to southeast Ecuador and northern Brazil in the Amazon region [31, 36]. In Colombia, both morphospecies *E. ruidum* sp. 1 and sp. 2 have been recorded [37, 38]. Regarding their feeding habits, although *Ectatomma* species are considered to be hunters, the *E. ruidum* complex is characterized by a very broad generalist diet that includes a great variety of prey, vegetable carbohydrates (fruits, extrafloral nectaries), honeydew produced by hemipterans, carcasses of both invertebrates and other animals, and seeds [27, 30, 39–41]. This species is a functional agent in Neotropical agroecosystems [30, 38, 39], exhibiting a high bioturbation impact during nest excavation due to high nest densities [6]. In Mexico, Costa Rica, and Panama, this species has been considered monodomous (each colony occupies a single nest structure) due to the architecture of its nests being characterized by only one nest per colony and each nest being an individual entity independent from other, neighboring nests (each nest contains at least one queen and shows territoriality against neighboring homospecific nests) [30, 42, 43]. However, the common occurrence of intraspecific cleptobiosis, that is, the theft of food resources between neighboring nests [42, 44, 45], and, on some occasions, the transport of workers and even brood between them [46], suggest the possible existence of a more complex nesting structure. More specifically, in southwestern Colombia,

this assertion is not at all clear given that various fragmentary observations [47–49] have reported quite frequent surface relationships between neighboring nests and the absence of queens in most of them, suggesting a polydomic nesting strategy (a colony occupies two or more spatially separated nests) [6, 28].

The purpose of this study was both to obtain basic information about the physical structure of the nest of *E. ruidum* sp. 2 by studying its nest architecture (structure of the entrance, number, volume, and depth of the chambers, depth of the nest), the composition of its colonies (number of queens, alate females (gynes), males, workers, pupae and larvae, and presence of possible myrmecophilic organisms), and the distribution of the individuals within the different chambers, and to try to correlate this information with the local nesting strategy (monodomy vs. polydomy). The aim of the study was to answer the following questions: (1) Does the nest architecture of *E. ruidum* sp. 2 show species-specific characteristics? (2) Are the colonies monogynous or polygynous and does the presence of the queen influence the physical and populational characteristics of the nests? (3) Does the nest architecture confirm the physical independence of neighboring nests and support a polydomic nesting strategy?

## Material and methods

### Ethics statement

N/A. Collection and transport of insect specimens were authorized by the Autoridad Nacional de Licencias Ambientales (Permit No. 1070 to the Universidad del Valle). The collection did not involve endangered or protected species. Research and fieldwork complied with the current laws of Colombia.

### Study areas and nest extraction

The Andean populations of *E. ruidum* sp. 2 from two locations in southwestern Colombia were studied. The first locality was in the municipality of Santiago de Cali (3°22'33.52" N, 76° 31'47.48" W), in the Department of Valle del Cauca, specifically in the Mélendez Campus of the Universidad del Valle (henceforth referred to as 'Cali'). The mean altitude and temperature are 974 m.a.s.l. and 24.7°C, respectively, with an annual rainfall of 1400 mm distributed in a bimodal pattern with two high peaks in April-May and October-November [50]. With a total area of 100 ha, this is a predominantly urban zone, with tree-lined areas and others without major tree cover. In a completely open 50 x 50 m plot (without bushes or trees) situated at the eastern end of the campus, all *E. ruidum* sp. 2 nests present were located through the use of bait and the tracking of foragers. The number of nests present in the area was recorded twice: between June and July 2016, coinciding with the dry season, and in January 2017, usually coinciding with a less drastic dry period during the transition season (period of transition from the rainy season to the dry season), although the climatic conditions were not respected because unusual rainfall for this period was recorded in January 2017. All the nests found in the first sampling were marked with flags (Fig 1A) and GPS and mapped with grids in the field. Subsequently, to avoid problems associated with the loss of flags, they were marked with plastic bottle caps, buried in the ground next to the nest opening, so they could be compared to those found in the following sampling. During the two last weeks of January 2017, a sample of 118 *E. ruidum* sp. 2 nests was extracted, from which 15, containing less than 10 workers and no sexual adults and brood, were considered abandoned and were not considered for subsequent analysis. For each nest, the number of chambers and the total nest depth were recorded.

For 24 of these randomly chosen nests, the extraction was performed using the paraffin wax technique [51]. In previous studies, materials such as molten zinc or lead, dental plaster, cement, paraffin wax, or even ice have been used [15, 16, 19, 51–53]. We used paraffin wax

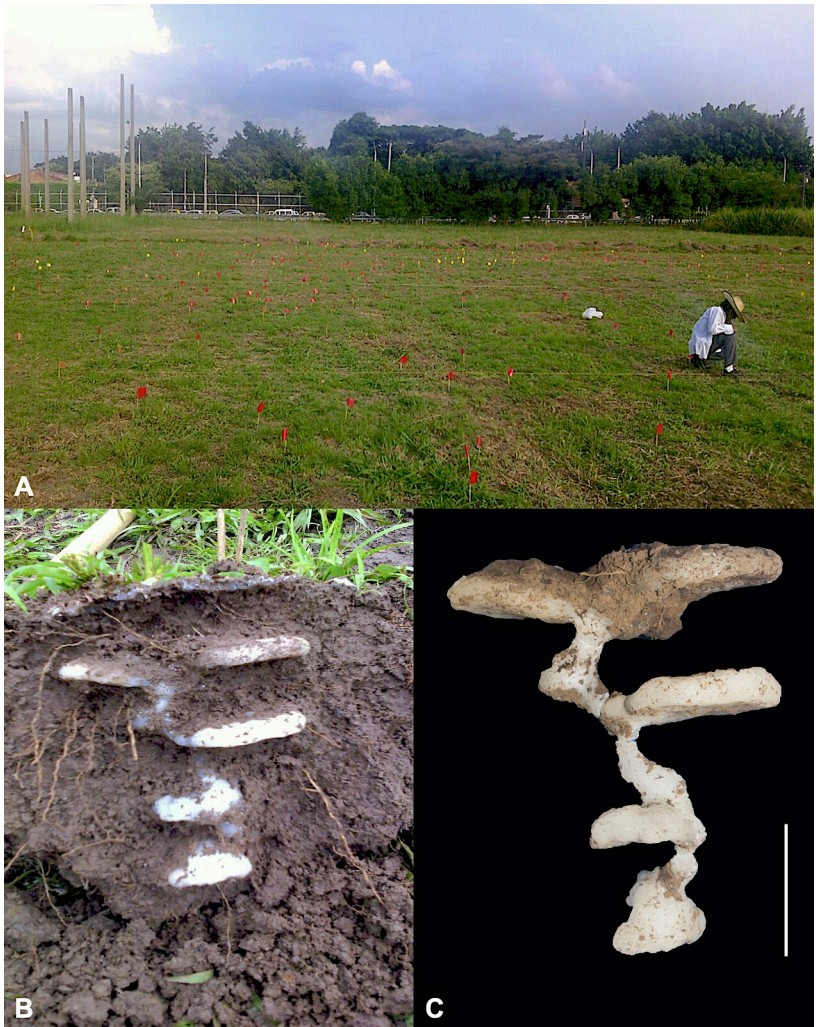

**Fig 1. Wax cast nest collection.** (A) study area at the Campus of Universidad del Valle, Cali, with the entrances of the nests marked by a flag, (B) wax casting of nest #8 embedded in its natural habitat prior to extraction, (C) the same nest #8 once cleared of its soil matrix; the scale bars correspond to 10 cm. Photos: C. Santamaría.

because it is an inexpensive material and is easy to obtain and handle. Primarily, it allows the composition of each chamber to be easily determined as, by melting the wax in each section, both the workers and the brood can be obtained intact [51]. The paraffin wax was poured into the entrance hole of each nest and left to dry for 24 hours; subsequently, each nest was extracted as gently as possible to prevent the paraffin from breaking (Fig 1B). Photographs were taken of the structure of the nests during the extraction process, ensuring that, if the nests were to disintegrate, their original structure could be recovered. Photographs of the nests cleared of their soil matrix were taken against a black background to obtain an enhanced definition of the contours in terms of nest shape and structure (Fig 1C). Using the geometric shape of structural elements in a similar way to that described for studying morphological transitions in the *Lasius niger* nest shape [54], we defined the start and end of both the tunnels and the chambers that were easily distinguishable because chambers were horizontal to the ground while the tunnel was rather vertical and clearly elongated. Chambers began at the point where the tunnel leading to them widened. Following extraction and photography, the

dimensions (length, width, height) of each chamber were measured with a millimeter ruler to obtain an estimate of the chamber volume; additionally, tunnel length and total depth of the nests were recorded. Then the nests were melted down chamber by chamber and the content of each one was recorded: number of queens, gynes, males, workers, pupae, and larvae, as well as the presence of other associated arthropods, if any.

The second locality was the Vereda El Rosal, in the southwestern region of the Colombian Andes, municipality of Caldono, Department of Cauca (2˚49' N, 76˚32' W) (henceforth referred to as 'Cauca'). The mean altitude and temperature are 1450 m.a.s.l. and 21.5˚C, respectively, with an annual rainfall of 2191 mm distributed in a bimodal pattern with two high peaks in April-May and October-November [55]. Between October and December 2016, 49 nests were extracted, with only the total depth of the nest recorded, and not the number of chambers.

## Data analysis

Pearson's correlation tests were performed to identify possible relationships between the numbers of workers, larvae, and pupae, and between chamber volume or group size (adults + brood) and nest depth. A Chi-squared test was performed to determine whether the proportion of queenright nests differed between both populations. A Binomial Generalized Linear Model (Binomial GLM) was performed to test for differences between the population composition of the excavated nests (number of larvae, pupae, workers, males, and gynes) and nest structure (nest depth) depending on whether or not a queen was present. Twelve exploratory models were first adjusted. The final chosen model used was as follows: glm (formula = Queens ~ locality + gynes + workers + larvae + depth + locality * male + locality * workers, family = binomial, data), with 'queen presence' as the random variable and using the library "car" [56].

To detect trends among all variables in nests with or without a queen, we performed a Principal Component Analysis (PCA) to determine which variables clustered with respect to queen presence. The explored variables were number of gynes, males, workers, larvae, and pupae and nest depth. We used the libraries "FactoMineR", "factoextra" and "corrplot". A separate test was conducted for all excavated nests and each locality. All statistical analyses were performed using R (version 4.1.1) [56], with a significance level of 5%.

## Results

### Nest abundance and general considerations on *E. ruidum* sp. 2 nest architecture

During the first nest sampling in June-July 2016, 457 nests of *E. ruidum* sp. 2 were recorded in the 50 x 50 m plot in 'Cali' (i.e., 1828 nests ha$^{-1}$). In contrast, in January 2017, 942 nests of this species were found in the same plot (i.e., 3768 nests ha$^{-1}$), with some of the former nests having disappeared, while many new nest entrances appeared. Although the difference between the first and second assessments was 485 nest entries, it was estimated that about 200 entries recorded during the first sampling had disappeared, making the number of new nests built between the two samplings actually 685. The average distance between neighboring nests was about 74 cm, indicative of the high nest density at this site. There was some worker traffic between nests, but we did not record an average per nest. However, we regularly observed workers transporting other individuals from one nest to another nearby.

All the *E. ruidum* sp. 2 nests extracted in this study were built without the presence of roots or plants, although in some cases they were found near the base of grass. Small supplementary

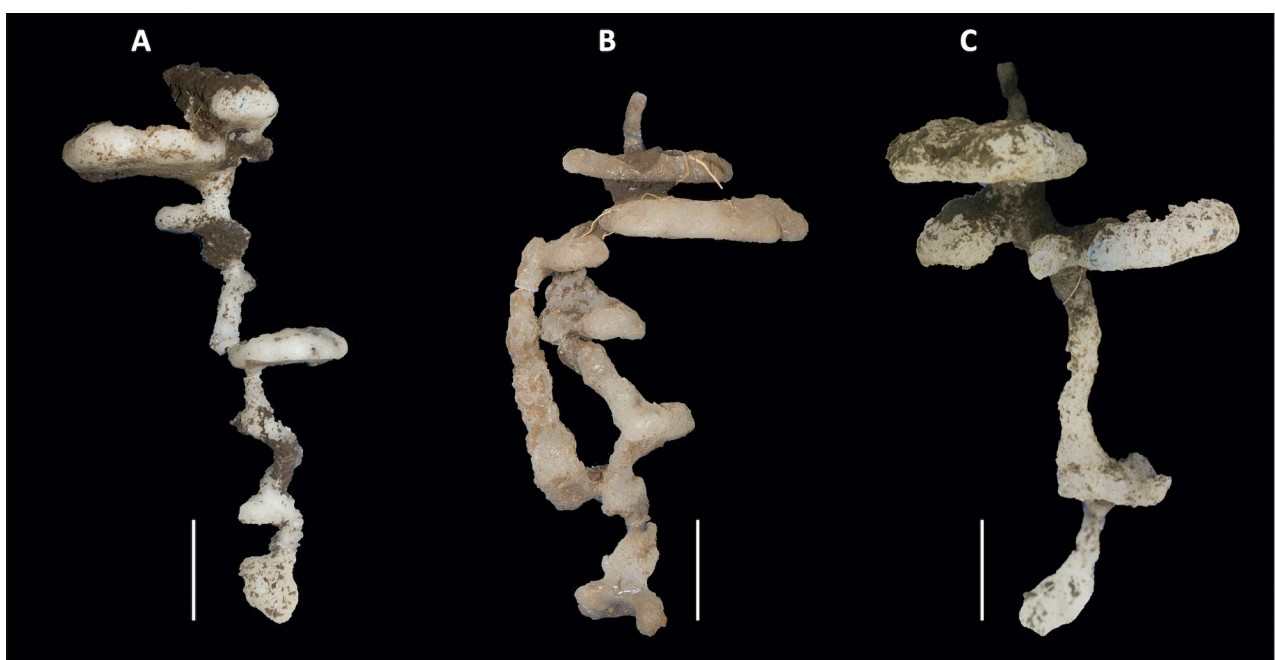

**Fig 2. Examples of nests of *Ectatomma ruidum* sp. 2 of similar depth, cast in paraffin wax.** (A) nest #3 (depth: 28 cm), (B) nest #6 (depth: 27 cm), (C) nest #22 (depth: 29 cm) (see S1 Table for detailed nest features). The scale bars correspond to 5 cm. Photos: C. Santamaría.

chambers with refuse material or stored food were not present in any of the nests. In particular, no surface or underground communication was observed using tunnels or galleries between the different nests, which all had the same isolated vertical arrangement, typical of the nests of this genus (Fig 2).

Although not quantified, some specimens of Nicoletiidae (Zygentoma) could be observed in most nests. However, no other associated organisms were found during this study.

## Nest casts

The nests had only one circular entrance/exit hole, with an average diameter of 4.2 ± 0.2 mm (mean ± SEM) (see S1 Table); no specific structure (cone, tumulus, or chimney) was observed. In all cases, both the access tunnel and the chambers connected through this single tunnel were vertically arranged (Fig 2). Only one of the nests extracted using the paraffin wax technique (nest #6, Fig 2B) presented a second vertical tunnel connecting two non-contiguous chambers.

No galleries were found interconnecting the chambers or tunnels of different nests, either above or below ground, and no small 'appendix' chambers containing waste or stored materials (seeds, food, or insect remains) were observed. The first chamber was very superficial, at a mean distance from the entrance of 6.7 ± 0.2 cm (Table 1), and all the nests were very shallow with a mean depth of 29.1 ± 0.8 cm (range: 22–36 cm) (see S1 Table). Spacing between the chambers was relatively constant with no significant differences in the length of the tunnels connecting contiguous chambers (Table 1).

The tunnel walls and both the walls and floors of the chambers were lined with a cement-like, very compact, thin surface, smooth in appearance. The shape of the chambers was a half ellipsoid, and the deeper the chamber, the smaller its volume and group size (adults + larvae + pupae) per chamber (Table 1 and S2 Table), as evidenced by a significant negative

**Table 1. Mean (± SEM) volume and depth of the chambers, tunnel length, and composition of the population of each chamber according to its relative position for the 24 *E. ruidum* sp. 2 nests extracted using the paraffin wax technique.** * The average values reported for the ninth chamber concerning the number of workers and gynes, must be considered with reservation because only one nest with nine chambers could be cast; they cannot be directly compared to the mean values found in the other chambers.

| Chamber | n | Volume (cm³) | Chamber depth (cm) | Tunnel length (cm) | Queen | Workers | Males | Gynes | Larvae | Pupae |
|---|---|---|---|---|---|---|---|---|---|---|
| #1 | 24 | 28.6 ± 2.6 | 6.7 ± 0.2 | 4.6 ± 1.1 | 0.1 ± 0.1 | 10.2 ± 2.6 | 1.2 ± 0.3 | 0.1 ± 0.1 | 1.9 ± 0.8 | 3.9 ± 1.4 |
| #2 | 24 | 23.0 ± 2.6 | 9.6 ± 0.4 | 3.0 ± 1.3 | - | 10.1 ± 2.0 | 0.9 ± 0.3 | 0.2 ± 0.1 | 1.3 ± 0.7 | 1.6 ± 0.8 |
| #3 | 24 | 16.0 ± 2.2 | 14.2 ± 0.7 | 4.7 ± 2.3 | - | 15.5 ± 3.3 | 0.8 ± 0.3 | 0.3 ± 0.1 | 1.5 ± 0.7 | 2.6 ± 1.1 |
| #4 | 22 | 11.0 ± 1.5 | 18.5 ± 0.7 | 5.1 ± 1.9 | - | 10.5 ± 3.3 | 0.1 ± 0.1 | 0.1 ± 0.1 | 0.9 ± 0.7 | 1.6 ± 0.9 |
| #5 | 21 | 9.3 ± 1.9 | 23.1 ± 0.8 | 4.8 ± 1.9 | - | 8.2 ± 1.8 | 0.1 ± 0.1 | 0.1 ± 0.1 | 0.1 ± 0.1 | 0.2 ± 0.2 |
| #6 | 17 | 5.8 ± 0.9 | 26.5 ± 0.9 | 4.2 ± 0.9 | - | 7.9 ± 1.8 | 0.1 ± 0.1 | 0.1 ± 0.1 | - | - |
| #7 | 10 | 4.6 ± 0.9 | 29.2 ± 0.8 | 3.7 ± 1.5 | - | 4.4 ± 1.6 | - | 0.6 ± 0.4 | - | - |
| #8 | 5 | 2.4 ± 1.0 | 31.6± 1.2 | 3.0 ± 0.7 | - | 5.0 ± 2.4 | - | 0.2 ± 0.2 | - | - |
| #9 | 1 | 2.2 | 33.8 | 5.0 | | 5.0 * | - | 4.0 * | - | - |

correlation between chamber volume and depth (R(148) = - 0.6112; p < 0.001; Fig 3A) and between group size per chamber and chamber depth (R(148) = - 0.2656; p = 0.001; Fig 3B). However, there was no significant relationship between the number of workers, larvae, or pupae found in each chamber and the volume of that chamber nor between the distribution of the larvae and pupae in the chambers and their developmental stage. The GLM binomial analysis indicated that the distribution of the intranidal worker population was significantly concentrated in the three upper chambers (F = 4.18, p < 0.0001), but not directly associated with the volume of each chamber because it was especially concentrated in the third chamber (Fig 4). Although no statistical test could be performed due to the large variation in the number of samples for each relative position of the chamber, a decreasing trend of the chamber volume was observed according to its relative position (see Table 1). The mean total volume of all the chambers was 92.2 ± 7.4 cm³ per nest (n = 24).

### Relationships between colony composition and architectural variables

Considering the set of the 152 complete nests collected, some highly significant positive correlations, such as those found between the total number of workers per nest and the total number of larvae (S1A Fig) or pupae (S1B Fig) or between these last two categories of brood (S1C Fig), provided a good description of both the temporal and colony-level life of the colonies and their success because the higher the number of workers, the higher the brood production. Only 33 queenright colonies were found, all monogynous (S1 and S3 Tables). Depending on whether or not a queen was present, several significant correlations between the different variables related to the composition of the intranidal population and the structure of the nest were highlighted by the binomial GLM analysis (Table 2) and pairwise comparisons (S3 Table).

The presence of a queen had an overall significant positive impact on the number of workers and larvae, and a negative one on the number of gynes (p < 0.00001, p < 0.02, and p < 0.03, respectively; Table 2). In a comparison between localities (Table 2 and S3 Table), there was a highly significant negative impact of the presence of a queen on the number of males in 'Cauca' and a significant positive one on the number of workers (p = 0.0003 and 0.02, respectively). Although nest depth in the 'Cali' sample (28.7 ± 0.5, n = 103, range: 19–38 cm) tended to be shallower than that in the 'Cauca' sample (35.4 ± 1.8, n = 49, range: 21–80 cm) (see S3 Table), the binomial GLM did not indicate a significant difference. However, the numbers of workers (65.4 ± 3.3, range: 10–190 vs. 99.1 ± 7.6, range: 21–212), larvae (25.6 ± 2. 1, range: 0–92 vs. 74.2 ± 6.8, range: 6–203), and pupae (11.5 ± 1.1, range: 0–51 vs. 57.0 ± 7.0,

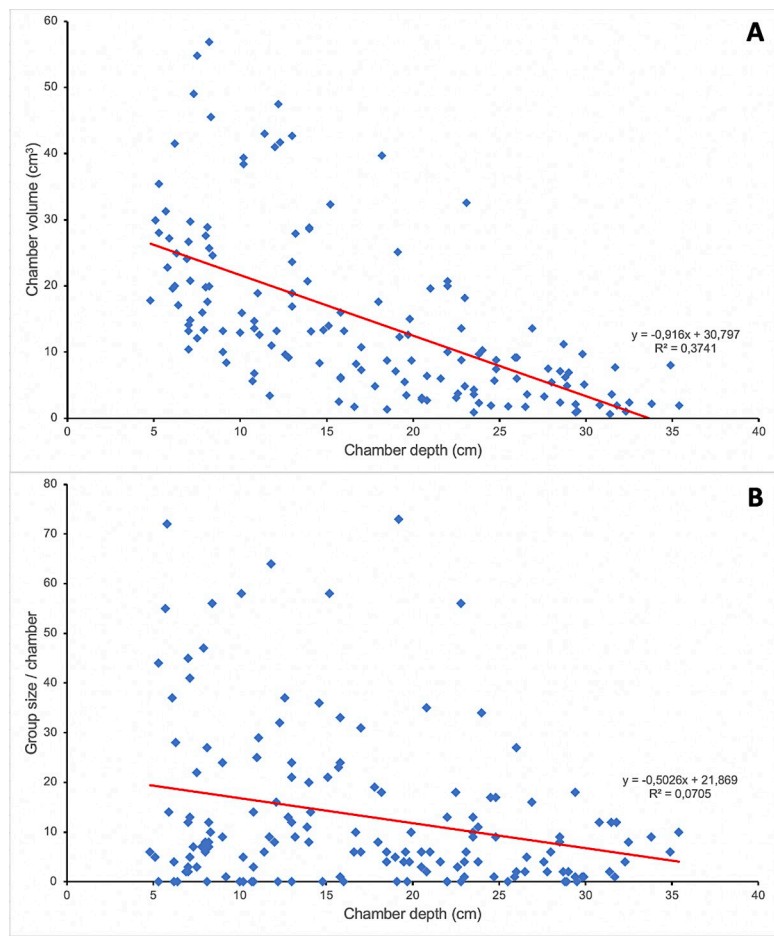

**Fig 3.** Correlation between chamber depth and (A) chamber volume or (B) group size per chamber. The deeper the chamber, the smaller its volume (R(148) = - 0.6112, p < 0.001) and the number of 'adults + brood' per chamber (R (148) = - 0.2656; p = 0.001).

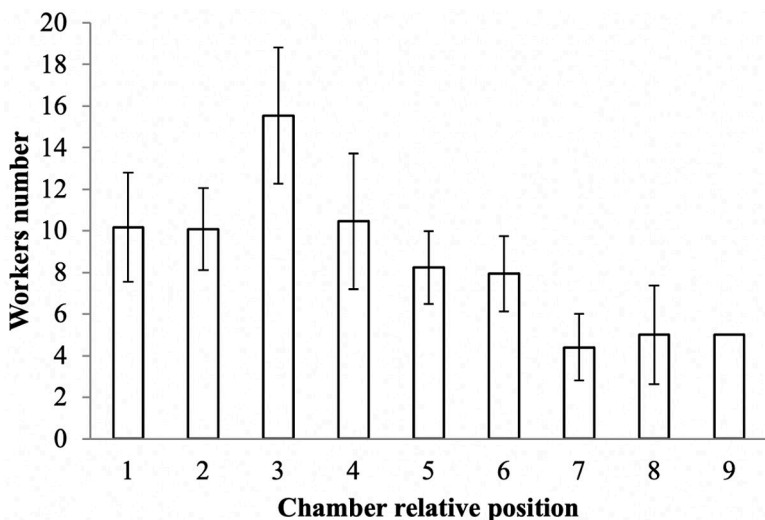

**Fig 4. Mean (± SEM) of *E. ruidum* sp. 2 workers per chamber in the 24 nests collected using the paraffin wax technique.** See Table 1 for the sample size and the depth corresponding to each chamber's relative position.

**Table 2. Binomial GLM comparing queenright vs. queenless nests.** Only significant results of the deviance analysis between population and structural variables of the 152 complete nests of *E. ruidum* sp. 2 extracted during the whole study are presented here. The number of workers and larvae was significantly higher in queenright nests while queenless nests had a significantly higher number of gynes. Queenright nests at 'Cauca' tended to have fewer males and more workers than at 'Cali'.

| | Chi-square | Df | Probability |
|---|---|---|---|
| Gynes | 47.072 | 1 | 0.0300357 |
| Workers | 315.728 | 1 | 0.921e-08 |
| Larvae | 53.687 | 1 | 0.0205009 |
| Locality: Males | 130.404 | 1 | 0.0003048 |
| Locality: Workers | 51.177 | 1 | 0.0236824 |

range: 0–194) were significantly lower in the 'Cali' sample than those in the 'Cauca' sample (see S1 and S3 Tables) as was the proportion of queenright nests (12/103 vs. 21/49; $\chi^2$ (1, $N = 152$) = 19.0244, p = 0.000013, S3 Table). In addition, the larva-to-worker and pupa-to-larva ratios (i.e., the number of larvae or pupae inside the nest divided by the number of workers inside the nest, respectively), both good indicators of colony growth efficiency, were significantly higher in the 'Cauca' sample than in the 'Cali' sample (0.75 and 0.77 vs. 0.39 and 0.45, respectively), indicating higher efficiency and a better growth rate in the 'Cauca' sample.

Overall, the PCA analysis (Fig 5) provided more details, with over 80% of the total variance in each of the three situations (all nests together, 'Cali' nests, and 'Cauca' nests) explained by the five main axes (dimensions 1 to 5 on the right plots) considered. In all three situations, the PCAs evidenced a strong positive impact of the presence of a queen on the number of workers and larvae and on the depth of the nests (Fig 5A–5C), but also a negative correlation between the number of gynes and males and the other variables. When all the nests were considered and at 'Cauca' (Fig 5A and 5C, respectively), the presence of a queen also had a strong impact on the number of pupae. When the data from 'Cali' were separated (Fig 5B), where the number of chambers per nest was recorded contrary to what occurred at 'Cauca', such an impact of the presence of a queen on the number of pupae was not observed but there was an additional strong positive impact on the number of chambers per nest, and the negative impact on the number of gynes was particularly clear.

Globally, the number of workers (135.1 ± 7.9 vs. 60.0 ± 2.3), larvae (86.8 ± 9.0 vs. 28.7 ± 2.1), and pupae (57.9 ± 9.3 vs. 17.4 ± 2. 1), as well as the depth of the nest (37.1 ± 2.3 vs. 29.1 ± 0.5), were significantly higher in queenright nests than in queenless ones (S3 Table). In contrast, in terms of the number of sexuals (gynes and males), the mean was higher in queenless nests as compared to queenright nests (0.7 ± 0.1 vs. 0.2 ± 0.1, and 1.7 ± 0.2 vs. 1.0 ± 0.5, respectively) (S3 Table).

## Discussion

### Nest distribution and nest architecture traits

Nest densities recorded in 'Cali' (1828 and 3768 nests ha$^{-1}$) tended to be higher than the range of densities previously reported in Colombia (360 to 3020 nests ha$^{-1}$ [40, 57]) and, in particular, that found in paddocks adjacent to 'Cauca', which had previously been estimated between 568 and 1945 nests ha$^{-1}$ [40]. The high rate of nest relocation and new entrance openings recorded in 'Cali' between the first and the second nest samplings (685 nests) was likely related to the action of the rain (unusual for this period of the year) in making the soil softer and easier to dig. It probably also resulted in some level of natural deterioration of the nests or of their quality, as suggested by the presence of 12.7% of abandoned nests among the 118 collected in

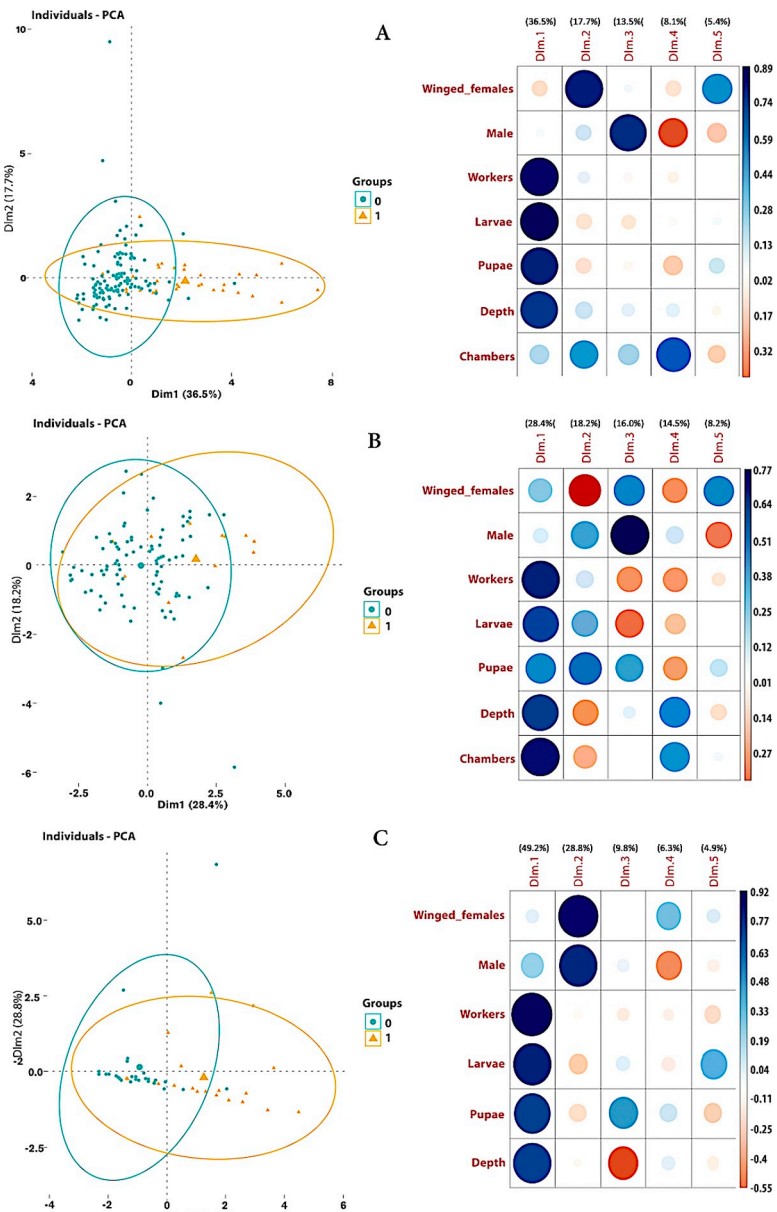

**Fig 5. Principal Component Analysis (PCA) including all structural and population variables of excavated nests according to the presence or absence of a queen.** Blue circles (0) represent queenless nests, and orange triangles (1) queenright nests. Dimensions 1 and 2 correspond to the two main axes of the PCA, with the highest eigenvalues, i.e. those that explained most of the variance in the system. (A) PCA using data from the 152 complete nests excavated; (B) PCA using the data from the 103 complete nests excavated at 'Cali'; (C) PCA using data from the 49 complete nests excavated at 'Cauca'. For each of the three plots at the right side of the figure, the size of the circles represents the amount of the effect of each dimension and each variable, while the intensity of their color represents the strength of the correlation. The blue color becomes more intense as the positive correlation gets stronger, while the red color becomes more intense as the negative correlation strengthens.

this plot, a rate over twice as high as that observed (5.6%) for the same species in southern Mexico [58]. Because only one collecting event was conducted in 'Cauca', and, furthermore, without prior sampling of nest distribution, it was not possible to disentangle the extent to which differences were the result of seasonality, site differences, or both. Nest relocation is a

common phenomenon in ants [59–61] and many hypotheses exist to explain it, including colony growth, intra- and inter-specific competition, nest deterioration, and seasonality. Although our data did not allow us to determine the exact causes of these high rates of nest relocation and new nest building, they did highlight that *E. ruidum* sp. 2 nest-building activity in this geographic area is apparently not limited to a period of only two months as recently suggested [6] but, rather, is spread over several months of the year. This could result in a significantly greater than expected soil bioturbation impact at the local level, well above the 0.9 to 3.1 kg of dry soil ha$^{-1}$ day$^{-1}$previously reported in Colombian Andean coffee plantations over two months of digging activity [6]. A conservative, rough estimate of the overall impact of bioturbation at 'Cali' could probably range from 2.9 to 6.0 kg of dry soil ha$^{-1}$ day$^{-1}$ throughout a nest building period longer than two months and could even reach 6.3 to 13.0 kg of dry soil ha$^{-1}$ day$^{-1}$ under more optimal conditions (see [6]).

In previous works that focused on *Ectatomma* species [20, 21, 23–26], the nest architecture in this genus has been described as simple, characterized by a single entrance/exit hole connected to a unique tunnel and a chamber system arranged vertically and occasionally associated with a lodge forming an appendix. In a few species (*E. opaciventre*, *E. brunneum* and *E. ruidum*), depending on either habitat characteristics or climatic conditions, entrances may be more elaborate, in the form of a chimney that generally consists of a mixture of soil and plant material [20, 24, 62]. In the *E. ruidum* nests described in Trinidad, Mexico, and Costa Rica [26, 30, 40, 42], the entrance hole is generally 3–5 mm in diameter, followed by a more or less vertical tunnel leading to 4–6 chambers, but occasionally up to 12 [62]. Our results showed that the general structure of the nests collected in southwestern Colombia followed the same pattern with a simple ground-level entrance/exit hole of approximately 4 mm in diameter, connecting to a vertical tunnel and an average of six chambers per nest, rather evenly spaced. None of the 152 excavated nests exhibited any external structure such as a cone, mound, or chimney of compacted earth, and none of them presented the structures that some authors call 'appendices', as reported for *E. planidens*, *E. brunneum*, and *E. vizottoi* [21, 23, 25], that contain colony remains and probably serve as nest refuse or could correspond to additional chambers under construction [23]. However, in nests of *E. ruidum* sp. 2 from Mexico [38, 46, 62], such auxiliary chambers were rarely observed and were usually empty, although we suspected that they were sometimes used to store garbage or food. Surprisingly, apart from the regular presence of silverfishes (Nicoletiidae) in the nests of *E. ruidum* sp. 2 collected during our study, the absence of any other associated organisms contrasted with previous studies that identified numerous associations with species of *Ectatomma*, in particular the *E. ruidum* complex, including *E. ruidum* sp. 2 in the same study area [47, 63, 64].

Comparable to several species of the genus, such as *E. opaciventre*, *E. planidens*, *E. brunneum*, and *E. vizottoi* [20–23, 25], the location of *E. ruidum* sp. 2 nests collected in this study was independent of the plants present in the habitat, confirming previous reports for this species in southeastern Mexico [65]. This characteristic contrasts with the case of *E. tuberculatum*, which, in almost all cases, builds its nests at the base of the trees on which it forages, closely associating the galleries and chambers with the roots of the vegetal (tree or shrub) support [29, 65, 66].

The internal cement-like surface lining the tunnel and chambers of *E. ruidum* sp. 2 was similar to the tightly-packed soil coating the inner walls of the chambers and the tunnels of *E. opaciventre* nests [20] and, as for this species, probably consisted of earth mixed with salivary secretions. It was similar to the clay-lining of the galleries and chambers used by the attine ant *Trachymyrmex turrifex* [67], the adobe-like lining of chambers in *Forelius chalybaeus* [68] and *Pogonomyrmex occidentalis* [69], or the mud-like material lining used by subterranean termites [70]. Such inorganic material lining probably helps regulate the internal climate of the nest by

preventing both humidity escape from the chambers and humidity entrance during rainfall [8, 67] or by stabilizing the tunnel and chamber walls [68, 71].

The mean total volume (92.2 cm$^3$) of the nests extracted with the paraffin wax technique corresponded relatively well with the mean values of soil removed per nest (between 104.4 g and 228.8 g) reported for *E. ruidum* sp. 2 in this geographic area [6]. The volume of the chambers was similar to that observed in *E. brunneum* nests [23], although this species is slightly larger in size than *E. ruidum* sp. 2. The significant negative correlation found in nest casts between the volume of the chambers and their depth was in contrast to the positive correlation reported for *E. opaciventre* and *E. brunneum*, for which chamber volume generally tended to increase accordingly with depth (see Table 1 in [20] and [23], respectively). At 'Cali', both the number of chambers and the depth of the nests of *E. ruidum* sp. 2 showed a highly significant positive correlation with the worker population size. Furthermore, similar to what has been reported for *E. opaciventre* and *E. brunneum* [20, 23], adult + brood population was concentrated in the chambers with the largest volumes (52.3 ± 7.6 vs. 29.9 ± 5.6 for all the other chambers) which, contrary to these species, however, corresponded to the upper three chambers rather than to the deepest ones. As in various other ground-dwelling ant species [11, 13, 72–74], field colonies of *E. ruidum* sp. 2 exhibited clear brood sorting with brood piles generally clustered in specific chambers. However, contrary to what is commonly observed in laboratory nests of this species (J.-P. Lachaud, unpubl. obs.), but similar to what has been reported in field nests of *Pogonomyrmex badius*, where larvae and pupae are not distributed differently from one another even though the vertical distribution of the brood is skewed toward the lower chambers [12], no clear relationship was found between the distribution of the larvae and pupae in the chambers and their developmental stage.

The most significant structural characteristic of the *E. ruidum* sp. 2 nests from southwestern Colombia concerns, in fact, the extreme shallowness of their depth. Their mean depth, even for the deepest ones found in the 'Cauca' sample (35.4 ± 1.8 cm), was considerably lower than the mean depth reported for *E. ruidum* sp. 2 in Mexico [38, 62], where they commonly reach 80–120 cm and, occasionally, up to two meters in the dry season [58, 62]. Such a difference in nest depth between southeastern Mexican populations and our Colombian populations of *E. ruidum* sp. 2 cannot be satisfactorily explained by a difference in the number of adults present in the nests, as the mean global number of adults per nest was quite similar (87.0 ± 2.4, n = 683 (J.-P. Lachaud, unpubl. data) vs. 78.7 ± 3.6, n = 152, respectively). A more likely explanation could involve the very different characteristics of soil structure in both habitats (the soil in the Mexican area where nest structure has been observed, near Tapachula in Chiapas, was comparatively very loose and essentially of volcanic origin (J.-P. Lachaud, unpubl. data)), or some local biological or ecological specificities.

## Queen influence on nest physical and populational characteristics and possible relationship between nest architecture and polydomy

The number of 33 queens obtained in the 152 *E. ruidum* sp. 2 nests extracted was extremely low, especially considering that Mexican and Costa Rican populations of this species are known to exhibit facultative polygyny [42, 75–78]. All queenright nests appeared to be monogynous and all queens were macrogynes, as were all other queens of *E. ruidum* sp. 2 found previously in the study area [47]. No ergatoid queens were found, though they were previously reported (as 'gynecoid') in this geographical zone [79], and no microgyne queens (small reproductive queens) were recorded, although they have been frequently reported in different Mexican populations of *E. ruidum* sp. 2 from Chiapas and Quintana Roo [28, 62, 76, 78].

The large predominance of queenless nests (78.3%) in our Colombian populations of *E. ruidum* sp. 2, and in particular at 'Cali' (88.4%), contrasts sharply with the situation in

southeastern Mexican populations, in which queenless colonies represented only between 8% and 11.3% of two samples of 408 and 203 complete colonies, respectively [58, 75]. As previously highlighted by Debout et al. [80], such a predominance of queenless nests, combined with the fact that gynes were significantly more numerous in these queenless nests, could indicate that this species is polydomous. Based on the total number of complete nests collected (152) and the number of queenright nests among them (33), one could argue that an average polydomous colony has 4.6 nests. However, this figure should be taken with caution and probably overestimates the actual number of satellite nests in our study area since previous results [58, 75] indicate that the absence of queens, at least in a fraction of the population, is a phenomenon that probably reflects the average natural mortality of colonies due to queen death or severe damage from heavy rains, as confirmed by the relatively high rate of abandoned nests (12.7%) reported in our study.

Queen presence in *E. ruidum* sp. 2 nests was positively correlated with both nest population size (number of workers, larvae, and pupae) and nest structure size (number of chambers and nest depth). A similar finding has been reported in the Spanish monogynous polydomic desert ant *Cataglyphis iberica*, in which queenright nests have significantly more workers than queenless nests and sexual brood is found only in queenless nests [81]. Such significant differences between queenright and queenless nests of *E. ruidum* sp. 2 could be related to the extreme shallowness of the Colombian nest in comparison to other known populations from Mexico, Costa Rica, and Panama and could account for the polydomic nesting strategy observed in the population studied [6, 47–49]. Nest size and, consequently, the number of chambers in which the nest population is segregated, may have important implications for the potential reproductive control by the queen because the fewer the chambers, the greater the diffusion of any regulatory pheromones released by the queen [82]. Some degree of functional inhibition, presumably of pheromonal origin, has been shown to occur in polygynous Mexican populations of *E. ruidum* sp. 2 [75, 77]. For example, in a study of a population of 226 nests, 42.1% of 83 polygynous colonies had only a single functional queen, although 94.4% of the 220 queens found in these polygynous colonies had been previously inseminated and would potentially have been capable of producing fertile eggs [77]. Such functional inhibition, confirmed later in another Mexican population using mitochondrial markers [78], may be better expressed in the Colombian nests due to their extreme shallowness. This hypothesis appears to be supported by the fact that at 'Cali', where the nests were significantly shallower than those at 'Cauca', both the number of queenright nests and the nest population size were significantly lower, while the presence of gynes was significantly higher than in queenright nests. Local soil constraints and the resulting nest shallowness could result in a more significant queen control in the studied Colombian populations, especially at 'Cali', and explain the necessity of nest splitting leading to the emergence of polydomy at the local level. In addition, high nest densities such as those observed in 'Cali' during this study, as well as the spatial aggregation of nests reported in some localities in southwestern Colombia [40], are probably also related to the expression of polydomy [80]; the annual cycle of this species would deserve to be studied in detail to verify the validity of such a hypothesis.

Overall, the studied Colombian populations of *E. ruidum* sp. 2 maintained the simple nest structure described for other species of this species complex from other geographical areas. However, our data suggest that *E. ruidum* sp. 2, at the local level, does not follow the usual monodomic pattern of this species with facultative polygyny but, rather, a polydomic pattern with monogyny, perhaps related to the extreme shallowness of the nests, which could significantly enhance the queen's reproductive inhibition previously reported for this species.

## Supporting information

**S1 Table. Composition of the population of the *E. ruidum* sp. 2 nests extracted at the University Campus of the Universidad del Valle ('Cali') and at Vereda El Rosal, Caldono ('Cauca').** Nests 1–24 were extracted using the paraffin wax technique. Q: queens; G: gynes; M: males; W: workers; P: pupae; L: larvae. The number of chambers was considered only for the nests collected at 'Cali' (103 nests). Nests 36, 44, 48, 52, 60, 72, 76, 81, 84, 88, 96, 99, 104, 106, and 114, containing fewer than 10 workers and no sexual adults and brood, were considered abandoned and were not considered for subsequent analysis. They are not reported in this table.
(PDF)

**S2 Table. Chamber dimensions (in cm), volume (in cm$^3$), and population within each chamber for the 24 *E. ruidum* sp. 2 nests extracted using the paraffin wax technique at 'Cali'.** Q: queens; G: gynes; M: males; W: workers; P: pupae; L: larvae.
(PDF)

**S3 Table. Mean (± SEM) nest population and nest depth for all collected nests according to the locality or the presence (queenright) or absence (queenless) of a queen, and at each locality according to the presence or absence of a queen.** 'Cali': Campus of the Universidad del Valle; 'Cauca': Vereda El Rosal, Caldano; n: nest sample size; Q: queens; G: gynes; M: males; W: workers; P: pupae; L: larvae. Student $t$ tests were performed to compare whether the presence or absence of a queen had any effect, with *: $p < 0.05$; **: $p < 0.01$; ***: $p < 0.001$; ****: $p < 0.0001$. #A Chi-square test was performed to compare the proportion of queenright nests in both localities with $\chi^2$ (1, $N = 152$) = 19.0244, p = 0.000013.
(PDF)

**S1 Fig. Relationship between different categories of nest population for the 152 complete nests of *E. ruidum* sp. 2. extracted.** (A) between the total number of workers per nest and the total number of larvae (R(152) = 0.6976, $p < 0.001$) or (B) the total number of pupae (R(152) = 0.6050, $p < 0.001$) and (C) between both categories of brood (R(152) = 0.7252, $p < 0.001$).
(PDF)

## Acknowledgments

We are indebted to Dagoberto Sinisterra, Omar Marín, Andrés López, Sebastián Narváez, Paula Palacios, Nicole Vargas, Bryan Ospina, Anderson Arenas, Sebastián Cañas and María Fernanda Rondón for their help during the hard work of nest excavation, to Julian Flavell for english revision, to Wilmar Torres, from The Universidad del Valle, Graduate Program in Biology, for statistical support, and to Nataly Amarillo for her help in designing Fig 5. We also thank four anonymous referees for helpful comments on a previous version of the manuscript.

## Author Contributions

**Conceptualization:** Inge Armbrecht, Jean-Paul Lachaud.

**Data curation:** Carlos Santamaría, Jean-Paul Lachaud.

**Formal analysis:** Carlos Santamaría, Jean-Paul Lachaud.

**Funding acquisition:** Inge Armbrecht.

**Investigation:** Carlos Santamaría.

**Methodology:** Carlos Santamaría, Inge Armbrecht, Jean-Paul Lachaud.

**Project administration:** Inge Armbrecht.

**Supervision:** Inge Armbrecht, Jean-Paul Lachaud.

**Validation:** Carlos Santamaría, Inge Armbrecht, Jean-Paul Lachaud.

**Visualization:** Carlos Santamaría, Jean-Paul Lachaud.

**Writing – original draft:** Carlos Santamaría.

**Writing – review & editing:** Carlos Santamaría, Inge Armbrecht, Jean-Paul Lachaud.

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
