## [Decision Letter · Decision Letter 0]

7 Jul 2021

PONE-D-21-13547

Nest architecture and colony composition in two populations of Ectatomma ruidum sp. 2 (E. ruidum species complex) in southwestern Colombia

PLOS ONE

Dear Dr. Lachaud,

Thank you for submitting your manuscript to PLOS ONE. After careful consideration, we feel that it has merit but does not fully meet PLOS ONE’s publication criteria as it currently stands. Therefore, we invite you to submit a revised version of the manuscript that addresses the points raised during the review process.

As can you see below, your paper was revised by four reviewers. After a careful reading of the manuscript, and the reviewers’ suggestions, my decision is a major revision of your paper. I recommend that you do a thorough review of English, and answer all the reviewers’ questions in detail. (Note that reviewer 1 has included an attachment with their comments)

Please, I would like you to know that the final acceptance of your manuscript will depend on the quality of the review of your manuscript and the responses to the reviewers' comments. Please let me know if you have any questions.

We look forward to receiving your revised manuscript.

Kind regards,

Maykon Passos Cristiano, D. Sc.

Academic Editor

PLOS ONE

Journal Requirements:

Reviewers' comments:

Reviewer's Responses to Questions

**Comments to the Author**

1. Is the manuscript technically sound, and do the data support the conclusions?

Reviewer #1: Partly

Reviewer #2: Yes

Reviewer #3: Yes

Reviewer #4: Partly

2. Has the statistical analysis been performed appropriately and rigorously? 

Reviewer #1: No

Reviewer #2: Yes

Reviewer #3: Yes

Reviewer #4: I Don't Know

3. Have the authors made all data underlying the findings in their manuscript fully available?

Reviewer #1: Yes

Reviewer #2: No

Reviewer #3: Yes

Reviewer #4: Yes

4. Is the manuscript presented in an intelligible fashion and written in standard English?

Reviewer #1: No

Reviewer #2: Yes

Reviewer #3: Yes

Reviewer #4: Yes

5. Review Comments to the Author

Reviewer #1: General comment

The paper brings information about the nest architecture of an ant species of economic importance in Colombia, which are relevant to this research field. However, the writing is very confusing and the applied analyses and their interpretation are very poor. It is necessary to reanalyze and rewrite the text. The discussion must be more reliant on your own results, instead of only existing literature.

Abstract

This section lacks an introduction with the theorical background that justifies your study. Also, you say in this section that you have strong correlations between the variables, but any correlation is near 90%. The conclusion here is much better than what is presented at the end of the discussion, but you need to write and present your data and analyses in a way that you can make the readers agree with you.

Introduction

The text of this section is too long and takes too long to approach the issue of the study. For example, the first paragraph has 27 lines and addresses different points concerning the nest architecture, such as how the construction material interferes with internal temperature, strategies to avoid flooding and also thermal preferences of individuals inside the nest. These issues are presented without any cohesion, making the text awkward. Also, the English needs improvement.

The second and third paragraph seem like a review and up to there I can not know what is the point of your study. And even with all these information, the importance/justificative of knowing the relation between nest architecture and population for these ant species is not presented.

Figure 1A is really very beautiful, but it does not illustrate the “high bioturbation impact during nest excavation” as is called in the text. The same is true for the figure caption “nest excavation”. What I see is an ant transporting a pellet of soil (?) with its mandibles. The same goes for the other photos of Figure 1. What do you want to show with these photos? The figure 1 is once again called for the images B, C and D at material and methods (line 151), but I still cannot see why the feeding habits are relevant to your study.

The most important justificative of your study is presented only after line 110, which is related to question: Are the nests monodomous or polidomous? On my point of view this is the main justificative of your study. I recommend you to shorten the text before this point and highlight this issue.

In the last paragraph, you first presented two objectives:

• to evaluate the spatio-temporal competitive relationships of their nests

• to validate, or not, the existence of a possible polydomy.

Following, you presented as objective what actually is your methodology. And then you presented some questions. The first one I can not understand at all. Please clarify. The others seem unrelated with your proposed objectives (listed above). Please clarify how answering these questions can help you reach your objective.

Material and Methods

Line 163-164. Why did you sample the nests in 2 different seasons? Furthermore, why did you choose these seasons? Please, justify.

Line 165. How did you mark the nests?

Line 166-170. You report that from 188 nests, 15 were not used as data due their small population. Following, you state that you molded 24 nests. What happened to the others? Please explain.

Line 179. Change “nests” for “chambers”.

Line 188-189. Why were the nests from Vereda El Rosal not molded?

Line 193. What exactly did you mean with chamber size? Chamber area? Chamber volume? Size is too vague. Please be more specific about what kind of data you analyzed.

Line 193-194. “Student-t tests were performed to compare whether the presence or absence of queens had any effect …” Effect of what over what?

Also, in the section Data Analysis you must tell the readers which was your random variable in the GLMM, which program and packages of statistical analyses you used. There are a lot of information that are lacking in this section.

Results

In general, the applied statistical analyses and specially their interpretations are not adequate (see detailed comments below). I strongly suggest that instead of doing a lot of paired correlations, you consider applying a multivariate analysis, in which you can verify which variables were associated to the similarities and dissimilarities between the nests. Even the locale can be incorporated in the multivariate analysis, making your result more robust. This way you can examine several variables (population parameters) to see if one or more of them are predictive of a certain outcome, in this case the nest architecture.

Line 242. “The tunnels walls and the floors of the chambers were lined with a very compact and thin surface …” This is very confusing to read and understand, please review the English writing.

Line 245-246. The negative correlation between chamber volume, not size, and depth does not mean that superficial chambers are larger. Instead, the analysis indicates that as the chamber was deeper, its volume was smaller. Also, the R value (-0.62) is too small. Please interpret the analysis according to the results it gives to you.

Lines 246-250. What is the MLGM analysis? It was not described in material and methods. Also, which chambers did you consider as the upper ones? The third chamber is not one of these upper chambers. Once again you can not say that the 3rd chamber was smaller than the 2nd or the 1st, as you did not apply any statistical test.

At Table 2, what does “nb” mean after the word worker or larvae? Here there is another statistical problem. Queen presence and population should not be analyzed with a linear regression, but instead with a logistic one, as presence/absence of the queen is a binomial variable.

Discussion

Just like the introduction section, the text is too long. The ideas are not well coordinated. Also, I can not find the answers to the proposed questions addressed at the end of the introduction. Especially in relation to the status of the species’ polidomy or monodomy. You suggested at the last line that they were probably monodomous, based in the fact that there are queenless nests. However, you did not find horizontal connecting tunnels among nests or any evidence of external communication. Thus, in which evidences is your answer based on? I recommend that while restructuring this section, you follow your proposed questions.

Reviewer #2: In this very interesting study authors have investigated the nest structure, colony composition and the overlap between these features in Ectatomma ruidum sp. Combining excavation with paraffin casts authors have studied a large number of colonies across two different populations at two different time points in Southwest Colombia. Having read the manuscript with enthusiasm as there are so few studies regarding this aspect and this part of the world, I have the following suggestions to improve the analysis and presentation.

• Introduction is well written and sets the stage for the questions is a nice manner. The one improvement required is the usage of certain terms. Abstract and introduction has certain terms like “monodomic” which researchers with a social insect background can understand, but for readers of this journal it would be useful if authors include the definition of these terms when it is used for the first time. I really like how line 31 reads, but if the meaning comes across to a non-specialist reader remains a question.

• Lines 141 to 151 needs to be moved away from the methods. This is background information about the study species and not related to how this study was performed. Further, the images mentioned are also not particularly relevant and the version with the current manuscript is not of sufficient resolution. As the current study is about nest structure and not the food or foraging of these ants its best to replace these images. An image of the nest entrance, image of the wax cast embedded in its natural habitat before the extraction and one image of the model organism would complement this manuscript in a better manner.

• Line 180 – Mentions that the different parameters of the nest was measured. How exactly was this performed? The tunnel and chamber would be hard to distinguish from each other. How do the authors understand which part of the nest is a chamber and which part is simply a blind extension of the tunnel or the tunnel itself? While the images of the wax cast indicate that the first couple of chambers are clear, how do the authors delineate the other chambers and measure the irregular structures. Other researchers have performed this in a quantitative manner for example see Bhattacharyya & Annagiri 2019.

• Lines 184 - Authors need to think critically if this second population of ants is really adding to the over all view of the manuscript. Getting in complete information for part of the time from 49 nests, may just be adding variation to the data without the benefit of enhancing the information about the species. I can understand that a lot of effort has gone into collecting this data and the authors can consider using this data at a later date with complete information.

• Lines 199: Can the abundance be reported per unit area and how this changed within each grid over time. From what is reported here we can only get a very rough idea. This kind of information would be very useful if it is done in a systematic manner.

• The organization of the results section needs to be improved. Abundance, Architecture, Relationships between the architectural elements and Relationship between colony composition and architectural elements are some of the sub-headings one can consider. Currently some results have a sub headings others do not, also “global set of nests” (line 252 ) is not a particularly meaningful sub heading.

• What was the relationship between colonies that were excavated and those that were cast? If they were performed in the same area at about the same time it would be very useful to see if the results from these two methods match with each other.

• Also, it would be useful if the authors end the nest architecture section with an overview of an average nest for this species in its natural habitat features across all the parameters examined.

• Were there any changes across the seasons in the architecture?

Reviewer #3: Dear editor, I appreciate the reminder of my name as reviewer of the paper “Nest architecture and colony composition in two populations of Ectatomma ruidum sp. 2 (E. ruidum species complex) in southwestern Colombia ”,and I hope to contribute to it. The authors show an important data about ant species nesting and colony structure. However, I can recommend the publication of this article after major revisions. I think that there are major flaw is in the methods because different number of colonies has been evaluated, and is not clear if the colonies was studied in a disturbed area since they basically used colonies sampled at the University. Further, the results do not bring to much novelty and should be explored in the discussion (see https://doi.org/10.1007/s00040-018-0666-z). Thus, I would recommend to accept after major revisions. Thanks, I enjoy read the paper.

I have read your manuscript carefully and I believe it is an interesting contribution. In overall the manuscript is reasonable; it is well written, the ideas are clearly presented, but I think that the figures and tables would be improved.

Specifically, I would recommend using the name of the species and explain that it could comprise a species complex, removing the sp. 2 from the text.

The limitations due the differences in sample size need to be addressed and better highlighted for readers.

Further, the discussion could be better explored if information and comparison with other species of taxa were considered in the discussion see the reference above as example.

I believe that these points must be addressed before it is ready to be published in Plos One.

Reviewer #4: This paper is based on a very large sample of ant nests. I commend the authors for that, and for the application of wax casting to the nest-census problem. They have collected a large amount of valuable data, but in my opinion, these data are under-analyzed. The story of this ant could be much more completely laid out.

Since the authors used the wax-casting method that recovers all nest contents in place, why do they report these data in such crude form in the main text (+, ++, +++, etc)? They report actual numbers as supplementary material, and do so by chamber, but under analyze these data in the main body of the paper. Most ants sort brood by stage, depth, conditions, etc. Indicative of the failure to see the importance of such details is line 255 where the authors refer to the correlations between worker numbers, pupae, larvae etc. as "very trivial." Rather than being trivial, they describe both the temporal and colony-level life of the colony and its success. The number of pupa in a colony is a measure of the current birth rate, the worker-larva ratio is a measure of efficiency, both predict the current growth rate. The density of workers is potentially an important social measure. If callows can be recognized, then this gives a crude measure of the age-structure of the nest population. I see these and many more as lost opportunities.

I don't see much value in publishing lists of correlations unless the lists include the slope and the R2 of the relationship. How much of the variance is associated with a factor, and how fast is the change in y as x changes, and in which direction? These are the relationships of interest. What is the increase in nest area (or volume) per worker? Brood per worker? Etc. Overall, I think the authors missed a meaningful analysis of the nest populations, focusing instead on minor details of nest architecture, both within and across populations.

While there is a comparison between collection sites, these confound season with site. The authors report site differences in the nest depth, the number of workers and brood, as well as in the rate of queenrightness. To what degree are differences the result of differences in season, site or both? Among the most interesting are differences in queen number. The predominance of queenless nests indicates that this species is polydomous, but polydomy can be seasonal as well as related to colony size. Some ant species rear sexuals in either the presence or absence of queens. I think these data can tell us something about the annual cycle of this species. The high nest density is probably also relevant to polydomy. What was the spacing between nests like? Was there worker traffic between them?

Commendably, the authors present much of their raw data as Supporting Information, but the data I see there should have been more thoroughly analyzed and presented as graphs and figures in the main text.

Some other points: Line 44 citations on bioturbation should include papers by Tschinkel on Pogonomyrmex and Trachymyrmex bioturbation. Then, after introducing bioturbation as a subject, the authors do not discuss how Ectotomma contributes to bioturbation in its habitat. Since they have data on nest volumes and their rates of abandonment and new appearance, they could conceivably comment on bioturbation.

Line 71-2, Tschinkel provided such data for several other species as well, and never made any claims about plant roots (Line 75). The authors might find his recently published book on Ant Architecture helpful. I think the Introduction needs to be briefer and more focused, and that is true for the discussion too.

Minor points: Line 201-203 doesn't really describe turnover in the literal sense. In turnover, the total numbers remain more or less stable, but the membership changes. Here we have disappearances matched with far more new nests. A different description is needed, as well as an interpretation of what the patterns may mean.

I think this could be made into a paper that reveals much more about the biology of this ant than it currently does. I hope the authors will succeed at doing this.

6. PLOS authors have the option to publish the peer review history of their article (what does this mean?). If published, this will include your full peer review and any attached files.

Reviewer #1: No

Reviewer #2: No

Reviewer #3: No

Reviewer #4: No

---

## [Author Response · Author response to Decision Letter 0]

19 Sep 2021

Reviewer #1: General comment

The paper brings information about the nest architecture of an ant species of economic importance in Colombia, which are relevant to this research field. However, the writing is very confusing and the applied analyses and their interpretation are very poor. It is necessary to reanalyze and rewrite the text. The discussion must be more reliant on your own results, instead of only existing literature.

R: Please, see our answers point-by-point below.

Abstract

This section lacks an introduction with the theorical background that justifies your study. R: This was added in the revised version (see lines 17-20).

Also, you say in this section that you have strong correlations between the variables, but any correlation is near 90% R: We don’t really understand what the problem is that the reviewer raised. Probably because our cast nest sample size was high enough, the fact is that the correlations we are presenting are significant even for relatively small R-values and that is what we are reporting.

The conclusion here is much better than what is presented at the end of the discussion, but you need to write and present your data and analyses in a way that you can make the readers agree with you R: We tried to focus the revised discussion more directly on our own results and to present more convincing arguments.

Introduction

The text of this section is too long and takes too long to approach the issue of the study. For example, the first paragraph has 27 lines and addresses different points concerning the nest architecture, such as how the construction material interferes with internal temperature, strategies to avoid flooding and also thermal preferences of individuals inside the nest. These issues are presented without any cohesion, making the text awkward. Also, the English needs improvement. R: We agree with the reviewer and have considerably reduced this first paragraph, removing a number of details that made the text cumbersome. We hope that now the sequence of arguments is easier to follow. On the other hand, with regard to the English, as noted in the acknowledgements and as has been our practice for the past seven years, this manuscript has been checked by an English-speaking colleague, Julian Flavell, whose quality of expression in his own language has so far never been questioned.

The second and third paragraph seem like a review and up to there I can not know what is the point of your study. R: The reviewer is right. The second paragraph and the first two sentences of the third paragraph have been removed to get more directly to the presentation of the issue of our study.

And even with all these information, the importance/justificative of knowing the relation between nest architecture and population for these ant species is not presented. R: We have added a sentence to introduce this topic just before addressing the architecture of the nests in the genus Ectatomma (see lines 57-59 in the revised version).

Figure 1A is really very beautiful, but it does not illustrate the “high bioturbation impact during nest excavation” as is called in the text. The same is true for the figure caption “nest excavation”. What I see is an ant transporting a pellet of soil (?) with its mandibles. The same goes for the other photos of Figure 1. What do you want to show with these photos? The figure 1 is once again called for the images B, C and D at material and methods (line 151), but I still cannot see why the feeding habits are relevant to your study. R: We agree that this figure was not really relevant for our study as noted also by Reviewer #2 but merely illustrative. It has been removed.

The most important justificative of your study is presented only after line 110, which is related to question: Are the nests monodomous or polidomous? On my point of view this is the main justificative of your study. I recommend you to shorten the text before this point and highlight this issue. R: As suggested, we shortened the introduction and provided more details to explain our objectives. 

In the last paragraph, you first presented two objectives:

• to evaluate the spatio-temporal competitive relationships of their nests

• to validate, or not, the existence of a possible polydomy.

Following, you presented as objective what actually is your methodology. And then you presented some questions. The first one I can not understand at all. Please clarify. The others seem unrelated with your proposed objectives (listed above). Please clarify how answering these questions can help you reach your objective. R: We agree with the reviewer that this paragraph lacked clarity and that there was some confusion and redundancy in the questions asked. We have therefore modified this paragraph by making it clearer (at least we hope so) which parameters were recorded and which questions were asked, avoiding redundancy (see lines 92-102 in the revised version).

Material and Methods

Line 163-164. Why did you sample the nests in 2 different seasons? Furthermore, why did you choose these seasons? Please, justify. R: Because in our manuscript we didn’t really analyze the effects of seasonality on nest architecture, and because seasonality is only marginally addressed when we talk about nest relocation, we decided to avoid mentioning seasons in this paragraph. However, we left the indication of the months in which the number of nest entrances were recorded and specified that, theoretically, the two sampling periods chosen were supposed to be during the dry season, although, beyond our control, the climatic conditions were not respected since unusual rainfall for this period were recorded in January 2017 (see lines 121-124 of the revised version).

Line 165. How did you mark the nests? R: Thanks for the comment; this is now specified in the revised version (see lines 124-128). Nests were marked with flags and GPS and mapped with grids in the field. Subsequently, to avoid problems associated with the loss of flags, they were marked with plastic bottle caps, buried in the ground next to the nest opening.

Line 166-170. You report that from 188 nests, 15 were not used as data due their small population. Following, you state that you molded 24 nests. What happened to the others? Please explain. R: It was not possible to cast all nests for logistical reasons. Furthermore, we consider 24 randomly chosen nests to be a representative sample of the population, and it should be noted that this sample size is similar or even larger than those typically used in this type of research. Finally, we had clearly stated, in lines 144-145 of the previous version before discussing the 24 nests that were used to make the casts, that: "For each nest, the number of chambers and the total nest depth were recorded". This information collected from the 79 nests on the University Campus in Cali that were not wax cast has been used in combination with that obtained for the 24 cast nests to calculate average values for the number of chambers and nest depth at the population level in ‘Cali’ (the complete data are presented in Table S2). In fact, as specified on line 143 of the previous version of our manuscript, 118 nests were extracted at ‘Cali’ (including the 15 incomplete ones), not 188 as reported by the reviewer.

Line 179. Change “nests” for “chambers”. R: Here, we are actually talking about "nests", not "chambers".

Line 188-189. Why were the nests from Vereda El Rosal not molded? R: Besides the fact that this technique takes a lot of time, the reason we didn’t cast the nests in Vereda El Rosal is that the coffee plots owners didn’t authorize pouring wax on their land, since these plots were productive and they feared that we would burn their soil. The difference with the University Campus is that, in this case, it was an experimental field and we were able to get the permits easily.

Line 193. What exactly did you mean with chamber size? Chamber area? Chamber volume? Size is too vague. Please be more specific about what kind of data you analyzed. R: We specified in the revised version (see lines 26, 94 and 149) that the parameter taken into account was the volume and we changed 'chamber size' by 'chamber volume' everywhere in the text where it was necessary.

Line 193-194. “Student-t tests were performed to compare whether the presence or absence of queens had any effect ...” Effect of what over what? R: Student-t tests were performed to compare whether the presence or absence of queens had any effect on the physical and populational characteristics of the nests. Given that we re-analyzed our data using a Binomial Generalized Linear Model, we changed this part of the “Data analysis” in the Material and Methods section (see lines 168-183 in the revised version).

Also, in the section Data Analysis you must tell the readers which was your random variable in the GLMM, which program and packages of statistical analyses you used. There are a lot of information that are lacking in this section. R: As noted in our previous response just above, we made changes in the “Data analysis” of the Material and Methods section of our revised version and provided the missing information.

Results

In general, the applied statistical analyses and specially their interpretations are not adequate (see detailed comments below). I strongly suggest that instead of doing a lot of paired correlations, you consider applying a multivariate analysis, in which you can verify which variables were associated to the similarities and dissimilarities between the nests. Even the locale can be incorporated in the multivariate analysis, making your result more robust. This way you can examine several variables (population parameters) to see if one or more of them are predictive of a certain outcome, in this case the nest architecture. R: We performed a binomial GLM but also a principal component analysis with our data to check whether different grouping would result according to the presence or absence of a queen. We change this part (see lines: 171-183 in the revised version).

Line 242. “The tunnels walls and the floors of the chambers were lined with a very compact and thin surface ...” This is very confusing to read and understand, please review the English writing. R: We are sorry, but we don’t understand what was not understandable in this sentence which, by the way, had already been checked by an English-speaking colleague, Julian Flavell. We are referring here to the inner lining of the chambers and tunnels which was very compact and smooth in appearance. We have only slightly modified the sentence to make it clearer (see lines 236-237 in the revised version). Such internal lining of chamber and tunnel walls with inorganic material has rarely been described in detail but is known in various soil-dwelling ants and termites. This is now specified and developed in the discussion of the revised version (see lines 375-383).

Line 245-246. The negative correlation between chamber volume, not size, and depth does not mean that superficial chambers are larger. Instead, the analysis indicates that as the chamber was deeper, its volume was smaller. R: To avoid confusion, we changed the formulation to: ‘the deeper the chamber, the smaller its volume’ and provide a new figure (Fig. 3 in the revised version) supporting this result (see lines 238-241). 

Also, the R value (-0.62) is too small. Please interpret the analysis according to the results it gives to you. R: We didn’t do anything else. Because of the large sample size of cast nests, the number of pairs compared (chamber volume vs. chamber depth) was hight (n = 148). As a consequence, although the R-value was small it was statistically highly significant (p < 0.001). We therefore concluded that there was a significant negative relationship beyond mere chance.

Lines 246-250. What is the MLGM analysis? It was not described in material and methods. R: We thank the reviewer for highlighting this error. The correct abbreviation was 'GLMM', which was described in the Material and Methods section. However, this remark is no longer relevant because this part has been modified in the new version using different methods of analysis (see "Data Analysis" in the Material and Methods section of the revised version). 

Also, which chambers did you consider as the upper ones? R: We now specify that we consider the three first chambers, closest to the surface, as the upper ones (see line 245 in the revised version). 

The third chamber is not one of these upper chambers. R: In fact, that’s just the case! As the mean depth of the 24 cast nests was 29.1 ± 0.8 cm (see Table S1) and the mean depth of the third chamber was 14.2 ± 0.7 cm (see Table 1), the third chamber was indeed one of the upper ones, located in the upper half of the nests, closest to the surface. 

Once again you can not say that the 3rd chamber was smaller than the 2nd or the 1st, as you did not apply any statistical test. R: We have rewritten this section and explained why a statistical test couldn’t be performed (see lines 244-250 in the revised version).

At Table 2, what does “nb” mean after the word worker or larvae? R: “nb” used to mean “number”, but this remark is no longer relevant because both Tables 2 and 3 presented in the previous version have now been removed. 

Here there is another statistical problem. Queen presence and population should not be analyzed with a linear regression, but instead with a logistic one, as presence/absence of the queen is a binomial variable. R: We hope that with the new analysis (Binomial GLM) this problem is now fixed (see lines: 171-177 in the new version).

Discussion

Just like the introduction section, the text is too long. The ideas are not well coordinated. Also, I can not find the answers to the proposed questions addressed at the end of the introduction. R: Changes have been made in the discussion and several additional details have been provided to explain our results and hypothesis. Consequently, the discussion was not really shortened, but we hope that, at least, it is now clearer. 

Especially in relation to the status of the species’ polidomy or monodomy. You suggested at the last line that they were probably monodomous, based in the fact that there are queenless nests. However, you did not find horizontal connecting tunnels among nests or any evidence of external communication. Thus, in which evidences is your answer based on? R: We suspect there was some confusion here between what is considered as ‘polydomy’ and ‘unicoloniality’. In the new version of the manuscript, we have made various changes and dedicated a special section of the discussion to present our evidences and hypothesis concerning the polydomic stucture observed in our Colombian populations (see lines 417-467). 

I recommend that while restructuring this section, you follow your proposed questions.

Reviewer #2: In this very interesting study authors have investigated the nest structure, colony composition and the overlap between these features in Ectatomma ruidum sp. Combining excavation with paraffin casts authors have studied a large number of colonies across two different populations at two different time points in Southwest Colombia. Having read the manuscript with enthusiasm as there are so few studies regarding this aspect and this part of the world, I have the following suggestions to improve the analysis and presentation.

• Introduction is well written and sets the stage for the questions is a nice manner. The one improvement required is the usage of certain terms. Abstract and introduction has certain terms like “monodomic” which researchers with a social insect background can understand, but for readers of this journal it would be useful if authors include the definition of these terms when it is used for the first time. I really like how line 31 reads, but if the meaning comes across to a non- specialist reader remains a question. R: Due to space limitations, we were unable to include the definitions of the terms "monodomic" and "polydomic" in the abstract, but we now provide a brief definition of these terms in the revised version when they are first used in the introduction (see lines 81 and 91).

• Lines 141 to 151 needs to be moved away from the methods. This is background information about the study species and not related to how this study was performed. R: Insofar as we had to significantly reduce our introduction to address the criticisms raised by reviewer #1 and get more directly to the presentation of our study problem, moving this paragraph into the introduction was somewhat counterproductive. However, we took into consideration the reviewer's suggestion and did so (see lines 68-78). 

Further, the images mentioned are also not particularly relevant and the version with the current manuscript is not of sufficient resolution. As the current study is about nest structure and not the food or foraging of these ants its best to replace these images. R: We agree that this figure was not really relevant for our study as noted also by Reviewer #1. It has been removed. 

An image of the nest entrance, image of the wax cast embedded in its natural habitat before the extraction and one image of the model organism would complement this manuscript in a better manner. R: We have attempted to address this comment by composing a new figure (Figure 1) that shows: 1) the collecting area at ‘Cali’, 2) the wax cast of a nest embedded in its natural habitat prior to extraction, and 3) the same nest once cleared of its earthy gangue. 

• Line 180 – Mentions that the different parameters of the nest was measured. How exactly was this performed? R: Measurements were made with a millimeter ruler. This precision has been added within the text (see line 149). 

The tunnel and chamber would be hard to distinguish from each other. R: In fact, the distinction between tunnel and chamber was less complicated than it seems because the two structures were clearly different: the chambers were horizontal to the ground whereas the tunnel was rather vertical and clearly elongated. We hope that the new Figure 1 will help the reader to distinguish the tunnel from the chambers. We think that if there was any confusion, it was extremely small and that the error was negligible. Chamber height was measured from the point where the tunnel that leads to it widens. Furthermore, because the same criterion was always used by the same researcher (CS), we are confident that the information is reliable. This information has been summarized and added to the revised version when describing the wax cast extraction (see lines 141-147).

How do the authors understand which part of the nest is a chamber and which part is simply a blind extension of the tunnel or the tunnel itself? While the images of the wax cast indicate that the first couple of chambers are clear, how do the authors delineate the other chambers and measure the irregular structures. Other researchers have performed this in a quantitative manner for example see Bhattacharyya & Annagiri 2019. R: Given it can sometimes be tricky to define, without ambiguity, where exactly a chamber started, we used the geometric shape of the tunnel or the chamber to determine the morphological transition of the structure and define their starting (or ending) point. In this way, the point where the tunnel begins to widen, and the wax begins to form an oval or circular shape was defined as a chamber. This is now explained in the revised version (see lines 141-147).

• Lines 184 - Authors need to think critically if this second population of ants is really adding to the over all view of the manuscript. Getting in complete information for part of the time from 49 nests, may just be adding variation to the data without the benefit of enhancing the information about the species. I can understand that a lot of effort has gone into collecting this data and the authors can consider using this data at a later date with complete information. R: We thank the reviewer for this suggestion, but we strongly believe that the information from the second study site (Vereda El Rosal) actually helps to strengthen the evidence we provide against our objectives. Indeed, the comparison of the average nest depth between the two studied areas and with the nests observed in Mexico, along with the significant difference in the proportion of queenright nests between ‘Cali’ and ‘Cauca’ that suggests a decrease in the expression of polydomy when nest depth increases, is essential for the hypothesis we proposed on the role of the queen's inhibitory control as a factor triggering polydomy (see lines 408-468 in the revised version). Therefore, we chose to retain these data.

• Lines 199: Can the abundance be reported per unit area and how this changed within each grid over time. From what is reported here we can only get a very rough idea. This kind of information would be very useful if it is done in a systematic manner. R: Because we didn’t initially plan to compare nest density or relocation over time and we only have two points recorded at different times, we can’t conclude on any trend over time. Therefore, we are unable to respond to this comment. We would need at least several records spread over longer periods of time to obtain a reliable pattern of nest relocation over time in a systematic manner.

• The organization of the results section needs to be improved. Abundance, Architecture, Relationships between the architectural elements and Relationship between colony composition and architectural elements are some of the sub-headings one can consider. Currently some results have a sub headings others do not, also “global set of nests” (line 252) is not a particularly meaningful sub heading. R: We agree with the reviewer, and reorganized the presentation of the Results section, separating the different parts with clear paragraph headings.

• What was the relationship between colonies that were excavated and those that were cast? If they were performed in the same area at about the same time it would be very useful to see if the results from these two methods match with each other. R: The colonies that were excavated in ‘Cali’ and those that were cast were from the same area and were extracted at the same time. Unfortunately, our results don’t allow to see if the results from these two methods match with each other because, for the colonies that were excavated, it was not possible to obtain the details of the internal composition of each chamber. First, because the workers didn’t remain still in the chambers due to disturbance during the excavation. Second, because workers couldn’t be collected chamber by chamber but in aggregate and as fast as possible due to logistical constraints and the number of nests to be collected.

• Also, it would be useful if the authors end the nest architecture section with an overview of an average nest for this species in its natural habitat features across all the parameters examined.

R: Considering that there is no previous report on E. ruidum´s nest architecture for Colombia, we think that this comment has been addressed, at least partially, in the “Discussion” section where we compared the known traits of nest architecture for E. ruidum colonies from different geographic areas (see lines 351-357 and 403-411 in the revised version).

• Were there any changes across the seasons in the architecture? R: That question is very interesting, but we don’t have the data to answer it, because we didn’t plan initially to study a possible seasonality effect on nest architecture and didn’t cast nests in ‘Cali’ over different seasons. Please, see also our response above to the comment of Reviever #1 on lines 163-164 of the previous version about a possible seasonality effect.

Reviewer #3: Dear editor, I appreciate the reminder of my name as reviewer of the paper “Nest architecture and colony composition in two populations of Ectatomma ruidum sp. 2 (E. ruidum species complex) in southwestern Colombia ”,and I hope to contribute to it. The authors show an important data about ant species nesting and colony structure. However, I can recommend the publication of this article after major revisions. I think that there are major flaw is in the methods because different number of colonies has been evaluated, and is not clear if the colonies was studied in a disturbed area since they basically used colonies sampled at the University. Further, the results do not bring to much novelty and should be explored in the discussion (see https://doi.org/10.1007/s00040-018-0666-z). Thus, I would recommend to accept after major revisions. Thanks, I enjoy read the paper. R: Thanks for the overall positive comment; however, we partially disagree with some of the arguments regarding the flaws due to differences in the number of colonies studied at the two sites and “disturbed area” status of our sites. First, we consider our research valid and novel because no previous studies on nest architecture had been made in Colombian populations of E. ruidum sp. 2, and for the first time we report evidences linking nest architecture and possible polydomy in this species. Regarding the comment on the ‘disturbed areas’, we must emphasize that both study sites were located in disturbed open areas. In the southwestern geographic area of Colombia, E. ruidum sp. 2 tends to colonize disturbed areas, and we assumed that the specificities of each habitat might induce different nesting strategies. The objective of our study was not to make a detailed comparison of the nests of this species between the two areas, but rather to determine if, due to a different nesting strategy, there might be variations in the population and nest structure. On the other hand, we are confident that, because the large sample size of nests at both sites allowed for statistically significant differences even when R-values were low, the difference in the total number of nests analyzed at the two sites can’t be a source of problem or bias.

I have read your manuscript carefully and I believe it is an interesting contribution. In overall the manuscript is reasonable; it is well written, the ideas are clearly presented, but I think that the figures and tables would be improved. 

Specifically, I would recommend using the name of the species and explain that it could comprise a species complex, removing the sp. 2 from the text. R: We are sorry to disagree, but we can’t remove the ‘sp. 2’ from the text. The fact that we are studying here the species sp. 2 of the E. ruidum complex is important since four other cryptic species have been distinguished until now in this complex, and various of them, presenting different biological traits, can be found in a same geographical zone. This has been reported for Mexico (see references 39–42, and 80 in the manuscript) and this is also the case for E. ruidum sp. 1 and sp. 2 that are present in Colombia (see ref. 39 and 41). We explain this situation in the presentation of the species in the introduction of the revised version (see lines 68-70 and 74-75).

The limitations due the differences in sample size need to be addressed and better highlighted for readers. R: As mentioned above, because of the large sample size at both study sites, there are no limitations due to the differences in sample size. The number of nests collected in both areas is more than sufficient for the analyses we performed, and, in fact, is significantly larger than most sample sizes reported in the literature and typically used in this type of study. We didn’t intend to statistically compare the two sites but rather to describe the characteristics of the nests at each site. Therefore, we are confident that this important information for both study sites is valid and worthy of presentation in the manuscript.

Further, the discussion could be better explored if information and comparison with other species of taxa were considered in the discussion see the reference above as example. 

I believe that these points must be addressed before it is ready to be published in Plos One. R: Both in the introduction and the dsicussion of the revised version of the manuscript, we are referring to various articles regarding nest architecture of other species of Ectatomma (see lines 59-67, 346-373, 385-395). Nevertheless, the focus of our study was E. ruidum sp. 2, and in our objectives we believe we have clearly specified that it is precisely the variation in nest architecture between populations of the same species, so far very poorly defined, that could provide insight into the factors triggering variations in nesting strategy.

Reviewer #4: This paper is based on a very large sample of ant nests. I commend the authors for that, and for the application of wax casting to the nest-census problem. They have collected a large amount of valuable data, but in my opinion, these data are under-analyzed. The story of this ant could be much more completely laid out. R: We thank the reviewer for this advice. We will take it into account in the near future, in a study more focused on the sociometry of the colonies, to understand better this ant species.

Since the authors used the wax-casting method that recovers all nest contents in place, why do they report these data in such crude form in the main text (+, ++, +++, etc)? R: The reviewer is perfectly right. It was our mistake to use (+, ++, +++, etc) in Table 1 while the complete detailed values were available in Table S2. This has been modified in the revised version (see new Table 1).

They report actual numbers as supplementary material, and do so by chamber, but under analyze these data in the main body of the paper. Most ants sort brood by stage, depth, conditions, etc. Indicative of the failure to see the importance of such details is line 255 where the authors refer to the correlations between worker numbers, pupae, larvae etc. as "very trivial." Rather than being trivial, they describe both the temporal and colony-level life of the colony and its success. The number of pupa in a colony is a measure of the current birth rate, the worker-larva ratio is a measure of efficiency, both predict the current growth rate. R: Thank you for this relevant and stimulating comment. Given that we completely changed the data analyses and used a Binomial GLM and PCAs for comparing queenright vs. queenless nests, the way the results were presented changed. However, when significant correlations were found, we tried to present them better, highlighting their relevance to the biology of the species (see Figs. 3 and S1 and lines 238-251, 258-263, 287-291).

The density of workers is potentially an important social measure. If callows can be recognized, then this gives a crude measure of the age-structure of the nest population. R: It is true that, as in many other ants, callow workers of Ectatomma have a distinctly lighter and softer cuticle than adults and, a priori, can be quite easily distinguished. Unfortunately, when melting the paraffin, due to a probably too high temperature, the workers were ‘cooked’ so that it was not possible to differentiate callows from adults as Tschinkel did for example with Pogonomyrmex badius (see ref. 16) and, therfore, the age-structure of the nest population couldn’t be assessed on the basis of the data presented here.

I see these and many more as lost opportunities. R: The reviewer is quite right, and we have obviously neglected to take into account a number of parameters that would have been important for a sociometric analysis of E. ruidum sp. 2 colonies. We can only admit that he is right in considering this lack of data as lost opportunities. However, we would like to emphasize that the essential objective of our work was not to perform a sociometric study of the Colombian populations of this species, but to investigate, at the level of the differences in nest architecture, which could be the factors explaining the use of a polydomic strategy at the local scale. 

I don't see much value in publishing lists of correlations unless the lists include the slope and the R2 of the relationship. How much of the variance is associated with a factor, and how fast is the change in y as x changes, and in which direction? These are the relationships of interest. What is the increase in nest area (or volume) per worker? Brood per worker? Etc. Overall, I think the authors missed a meaningful analysis of the nest populations, focusing instead on minor details of nest architecture, both within and across populations. R: The reviewer is correct. However, we probably don't need to add the R2 slopes anymore because the analyses have changed. In fact, the new statistical analyses are more robust and help to better understand how the population and nest structure variables change with the presence/absence of a queen. On the other hand, as stated slightly above, although our main objective was not a sociometric study of the Colombian populations of E. ruidum sp. 2, when significant correlations were found, we have tried to present them with emphasis on their relevance to the biology of the species. In view of the various comments of the reviewer on this subject, we are convinced not only of the usefulness of reprocessing all these data but especially of the need to complete them to prepare, in what we hope will be a not too distant future, a publication centered on the sociometry of E. ruidum sp. 2 colonies. However, this requires working conditions that the current pandemic situation doesn’t yet allow.

While there is a comparison between collection sites, these confound season with site. The authors report site differences in the nest depth, the number of workers and brood, as well as in the rate of queenrightness. To what degree are differences the result of differences in season, site or both? R: The reviewer is correct; we can’t discard a simultaneous effect of stationality and site. However, the exact way in which each of these factors affect nest architecture, that is, the contribution of each one separately, can’t be distinguished here. Therefore, we can’t conclude or try to compare both sites. We addressed this point in the revised discussion of our manuscript (see lines 335-337).

Among the most interesting are differences in queen number. The predominance of queenless nests indicates that this species is polydomous, but polydomy can be seasonal as well as related to colony size. Some ant species rear sexuals in either the presence or absence of queens. I think these data can tell us something about the annual cycle of this species. R: This is an interesting point. For the moment, we don’t have the data to verify if polydomy is seasonal or related to the size of the colony. However, the influence of colony size seems unlikely because, as noted in the discussion of the revised version (see lines 410-412) the overall average number of adults per nest reported for monodomous populations of E. ruidum sp. 2 in southern Mexico (87.0 ± 2.4, n = 683) is quite similar to that found in our polydomous Colombian populations (78.7 ± 3.6, n = 152). A study following the annual cycle of the species would deserve to be done.

The high nest density is probably also relevant to polydomy. R: Although we don’t have a definitive, precise answer regarding the influence of nest density on polydomy, we have briefly addressed this point in the revised discussion (see lines 457-461).

What was the spacing between nests like? Was there worker traffic between them? R: The average distance between neighboring nests was about 74 cm and a few individual transports were observed between nests. We have added this information in the presentation of the results in the revised version. (see lines 193-195).

Commendably, the authors present much of their raw data as Supporting Information, but the data I see there should have been more thoroughly analyzed and presented as graphs and figures in the main text. R: We performed new analyses (binomial GLM and PCA), removed unnecessary tables, and presented new figures both in the main text (Figures 3 and 5) and in the Supplementary Information (Figure S1) that hopefully provide more useful information.

Some other points: Line 44 citations on bioturbation should include papers by Tschinkel on Pogonomyrmex and Trachymyrmex bioturbation. R: Thanks for the suggestion. This was done. (see lines 44-45 and -497).

Then, after introducing bioturbation as a subject, the authors do not discuss how Ectotomma contributes to bioturbation in its habitat. Since they have data on nest volumes and their rates of abandonment and new appearance, they could conceivably comment on bioturbation. R: The reviewer is right. In fact, the bioturbation impact of E. ruidum sp. 2 nests has been addressed in a previously published paper and our objective here was not to focus on that aspect. We therefore simply refer merely to our publication in Pedobiologia (Ref. 9 in the manuscript) and only provide some complementary information resulting from our new data in the discussion of the revised version (see lines 340-346).

Line 71-2, Tschinkel provided such data for several other species as well, and never made any claims about plant roots (Line 75). R: While it is true that such a mention for Pogonomyrmex badius didn’t appear in Tschinkel's papers, in contrast in the last part of the discussion of the paper on Formica pallidefulva (Mikheyev & Tschinkel 2004), the authors clearly stated "Second, the high concentration of roots in the upper soil regions may provide structural support for the chambers, allowing the ants to build large chambers without risking cave-ins". In any case, this clarification is no longer relevant here since this part has been removed in the revised version.

The authors might find his recently published book on Ant Architecture helpful. R: Thank you for the suggestion. The recent book of Tschinkel was very interesting for a comprehensive view of nest architecture in ants and we quoted it in the introduction when presenting the influence of the nest architecture on both the spatial structuration of the colony population within the nest and their collective behavior (see lines 57-59 and 539-540). However, we didn’t find other new specific information useful for our discussion.

I think the Introduction needs to be briefer and more focused, and that is true for the discussion too. R: This has been made for the introduction. Regarding the discussion, some changes have been made in the presentation and several additional details have been provided to explain our results and hypothesis. Consequently, the discussion was not really shortened, but we hope that, at least, it is now clearer. 

Minor points: Line 201-203 doesn't really describe turnover in the literal sense. In turnover, the total numbers remain more or less stable, but the membership changes. Here we have disappearances matched with far more new nests. A different description is needed, as well as an interpretation of what the patterns may mean. R: The sentence was modified as follows “Although the difference between the first and second assessments was 485 nest entries, it was estimated that about 200 entries recorded during the dry season had disappeared, making the number of new nests built between the two periods actually 685” (see lines 190-193). Additionally, an attempt to interpret this pattern is now presented in the revised discussion (see lines 330-336).

I think this could be made into a paper that reveals much more about the biology of this ant than it currently does. I hope the authors will succeed at doing this. R: Thanks for the suggestion. Although it was not the aim of the present study, we have attempted to reveal more about ant biology, and we hope that the new version has succeeded better. However, future work focusing on sociometric analysis of colonies would definitely be needed.

---

## [Decision Letter · Decision Letter 1]

15 Nov 2021

PONE-D-21-13547R1

Nest architecture and colony composition in two populations of Ectatomma ruidum sp. 2 (E. ruidum species complex) in southwestern Colombia

PLOS ONE

Dear Dr. Lachaud,

Thank you for submitting your manuscript to PLOS ONE. After careful consideration, we have decided that your manuscript does not meet our criteria for publication and must therefore be rejected.

I forwarded the manuscript to the reviewers again and received input from two reviewers. After consideration of all revisions, I have decided that your manuscript does not reach scientific soundness for publications.

I am sorry that we cannot be more positive on this occasion, but hope that you appreciate the reasons for this decision.

Yours sincerely,

Maykon Passos Cristiano, D. Sc.

Academic Editor

PLOS ONE

Reviewers' comments:

Reviewer's Responses to Questions

**Comments to the Author**

1. If the authors have adequately addressed your comments raised in a previous round of review and you feel that this manuscript is now acceptable for publication, you may indicate that here to bypass the “Comments to the Author” section, enter your conflict of interest statement in the “Confidential to Editor” section, and submit your "Accept" recommendation.

Reviewer #3: All comments have been addressed

Reviewer #4: All comments have been addressed

2. Is the manuscript technically sound, and do the data support the conclusions?

Reviewer #3: Yes

Reviewer #4: Yes

3. Has the statistical analysis been performed appropriately and rigorously? 

Reviewer #3: Yes

Reviewer #4: Yes

4. Have the authors made all data underlying the findings in their manuscript fully available?

Reviewer #3: Yes

Reviewer #4: Yes

5. Is the manuscript presented in an intelligible fashion and written in standard English?

Reviewer #3: Yes

Reviewer #4: Yes

6. Review Comments to the Author

Reviewer #3: Dear authors,

The manuscript was improved and the majority of questions and suggestions has been explained or corrected in the main text. Now the mansucript can be accepted for publication.

Reviewer #4: The authors have greatly improved this paper since the first review. I understand that they did not always have the data to follow some of my suggestions, but they did a good job in bringing the message across. I have very few suggestions for improvement.

I had to look up the meaning of gangue. I don't think I would have used the word in that context, but it is colorful. I doubt it is often used outside the context of mining, but then, you never know how many miners will read this paper.

Line 265. This calculates to 4.6 nests per queen. Couldn't you claim that the average polydomous colony has 4.6 nests? Or do you think some colonies are truly queenless? You even state that there are no gamergates or ergatoid queens, so doesn't it follow that there are 4.6 nests per polydomous colony? Obviously, it would be nice to know the range, but at least you have a mean.

Line 346 and 388. Why not make an estimate of the amount of soil excavated by your population, especially since you did a second count and discovered many new nests? The mean weight and depth of the bioturbation rate could be estimated.

There are too many references. I suggest culling the list to be more targeted and relevant to the subject. This is not a review article.

7. PLOS authors have the option to publish the peer review history of their article (what does this mean?). If published, this will include your full peer review and any attached files.

Reviewer #3: No

Reviewer #4: No

- - - - -

---

## [Author Response · Author response to Decision Letter 1]

24 Nov 2021

Reviewer #3: Dear authors,

The manuscript was improved and the majority of questions and suggestions has been explained or corrected in the main text. Now the mansucript can be accepted for publication.

Response: We thank the reviewer. The comment is sufficiently explicit and clear and does not require any specific response from us.

Reviewer #4: The authors have greatly improved this paper since the first review. I understand that they did not always have the data to follow some of my suggestions, but they did a good job in bringing the message across. I have very few suggestions for improvement.

I had to look up the meaning of gangue. I don't think I would have used the word in that context, but it is colorful. I doubt it is often used outside the context of mining, but then, you never know how many miners will read this paper.

Response: We particularly appreciated the gently mocking tone of this comment. Not being English speakers ourselves, we followed the advice of the English colleague who revised our text, but we have to admit that the term "gangue" is perhaps too specific to mining, even if this term is perfectly appropriate in our respective languages of origin (Spanish and French). In our revision, we have replaced "earthy gangue " with "soil matrix " which is perhaps more commonly used and understandable to English speakers (see lines 142 and 157 in the new revised version).

Line 265. This calculates to 4.6 nests per queen. Couldn't you claim that the average polydomous colony has 4.6 nests? Or do you think some colonies are truly queenless? You even state that there are no gamergates or ergatoid queens, so doesn't it follow that there are 4.6 nests per polydomous colony? Obviously, it would be nice to know the range, but at least you have a mean.

Response: It is true that, based on our data, we can propose that an average polydomous colony has 4.6 nests even though the precise range is difficult to specify. However, as indicated in manuscript PONE-D-21-13547R1 (see lines 432-435 of the previous version), at least some of these queenless nests truly reflect a natural phenomenon of queen death or colony weakening. We have added a few lines in the discussion to take this comment into account (see lines 441-448 in the new revised version).

Line 346 and 388. Why not make an estimate of the amount of soil excavated by your population, especially since you did a second count and discovered many new nests? The mean weight and depth of the bioturbation rate could be estimated.

Response: The reviewer is absolutely right. Therefore, we have added in the discussion the requested bioturbation estimate (in kg of dry soil per hectare per day) taking into account the available values on our population (see lines 346-351 of the new revised version). 

There are too many references. I suggest culling the list to be more targeted and relevant to the subject. This is not a review article.

Response: We have done our best to address this comment by reducing the total number of citations by over 20%.

---

## [Editor Report · Decision Letter 2]

19 Jan 2022

Nest architecture and colony composition in two populations of Ectatomma ruidum sp. 2 (E. ruidum species complex) in southwestern Colombia

PONE-D-21-13547R2

Dear Dr. Lachaud,

We’re pleased to inform you that your manuscript has been judged scientifically suitable for publication and will be formally accepted for publication once it meets all outstanding technical requirements.

Kind regards,

Nicolas Chaline

Academic Editor

PLOS ONE
---

## [Editor Report · Acceptance letter]

24 Jan 2022

PONE-D-21-13547R2 

Nest architecture and colony composition in two populations of *Ectatomma ruidum* sp. 2 (*E. ruidum* species complex) in southwestern Colombia 

Dear Dr. Lachaud:

I'm pleased to inform you that your manuscript has been deemed suitable for publication in PLOS ONE. Congratulations! Your manuscript is now with our production department. 

Kind regards, 

on behalf of

Professor Nicolas Chaline 

Academic Editor

PLOS ONE